# Differentially Private Model Compression

**Fatemehsadat Mireshghallah**[1], **Arturs Backurs**[2], **Huseyin A. Inan**[2],
**Lukas Wutschitz**[3], **Janardhan Kulkarni**[2]
[1] University of California San Diego, [2] Microsoft Research, [3] Microsoft
`fmireshg@eng.ucsd.com`
`{arturs.backurs,huseyin.inan,lukas.wutschitz,jakul}@microsoft.com`

## Abstract

Recent papers have shown that large pre-trained language models (LLMs) such as BERT, GPT-2 can be fine-tuned on *private data* to achieve performance comparable to non-private models for many downstream Natural Language Processing (NLP) tasks while simultaneously guaranteeing differential privacy. The inference cost of these models – which consist of hundreds of millions of parameters – however, can be prohibitively large. Hence, often in practice, LLMs are *compressed* before they are deployed in specific applications. In this paper, we initiate the study of differentially private model compression and propose frameworks for achieving 50% sparsity levels while maintaining nearly full performance. We demonstrate these ideas on standard GLUE benchmarks using BERT models, setting benchmarks for future research on this topic.

## 1 Introduction

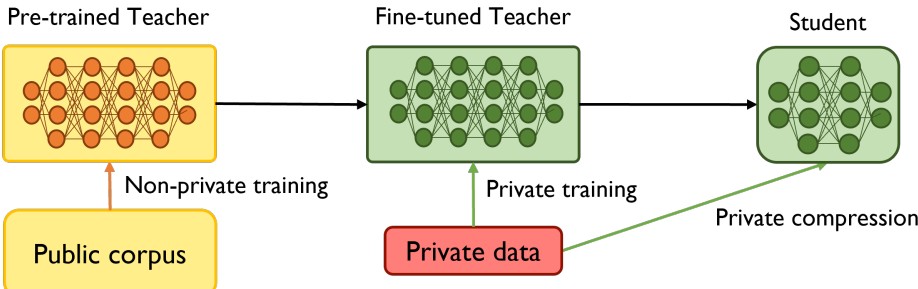

Figure 1: The 3-pronged modern deep learning pipeline: Pre-train on public data, fine-tune on private data, and compress the model to meet the memory and latency requirements of specific applications.

Since the advent of Transformer-based [67] large language models (LLMs) such as BERT [13, 37], GPT families [50, 5], there has been a paradigm shift in the way deep learning models are trained for natural language processing (NLP) applications. LLMs are first *pre-trained* on extremely large and diverse publicly available datasets. The weights are then *fine-tuned* for a specific task of interest using a much smaller *private dataset*, which may consist of sensitive information about the users. Finally, before deploying, the models are compressed (also referred to as sparsified or distilled) to reduce the parameter count. The reason for this final step is that LLMs such as BERT and GPT-2 consist of hundreds of millions of parameters, and hence inference time and memory footprints of the models are too large to be used in many applications. As a concrete example, consider a language model that is deployed for sentence completion task in text editors. In this scenario, clearly, the inference (response) time of the model needs to be in orders of milliseconds to be useful. Besides the

36th Conference on Neural Information Processing Systems (NeurIPS 2022).

practical considerations, it is also commonly observed that large deep learning models (even the ones that are not pre-trained) consist of many redundant parameters in the network that can be removed while retaining the full performance [32, 27, 26]. Hence most modern deep learning pipelines use this three-pronged approach of pre-train, fine-tune, and compress as illustrated in Figure 1.

While transformer-based models swayed deep learning research towards making models larger to achieve better performance, several recent works have shown that over-parameterized large models also increase the risk of leaking sensitive information about their training datasets [53, 8, 9]. Over the past few years, training deep learning models guaranteeing differential privacy (DP) [17], a strong notion of data privacy, has emerged as the defacto method to mitigate such information leakage. Exciting recent works have also shown that large pre-trained models when fine-tuned via DPSGD are as good as non-private models for a variety of NLP and image applications [75, 34, 12, 42, 44].

The main purpose of this paper is to understand how private training impacts the modern deep learning pipeline of pre-train, fine-tune, and compress. The main observation is that most widely used model compression algorithms such as Knowledge Distillation (KD) and Pruning, use private datasets to produce compressed models, hence if our goal is to deploy a differentially private compressed model, we should consider their impact on the training process. This leads to the question:

*What algorithms should one use to produce compressed private models and how do they impact private fine-tuning via DPSGD?*

The goal of this paper is to investigate this question and propose frameworks for private model compression in the context of NLP applications. Although we investigate model compression techniques at the fine-tuning stage using pre-trained models, we would like to emphasize that our frameworks for private model compression are not tied to this setting. They are equally applicable to training deep learning models from scratch and to other application domains such as image classification tasks.

## 1.1 Our Contributions

- We give a framework for doing model compression using Knowledge Distillation algorithm guaranteeing differential privacy, which we call DPKD. We show that DPKD alone is not enough to transfer the knowledge from large models to compressed models. This loss in the accuracy of compressed models can be mitigated by better initialization of compressed models from the weights of the large models, which itself is a form of knowledge transfer. We propose several *zero-shot, fully private* methods for initialization compressed models using weights of the large models. Our empirical evaluation of these ideas on standard GLUE benchmarks using BERT models show that DPKD approach to the model compression loses an accuracy of $5\%$ compared to the larger models if the compressed model has *half the size of the full BERT model*.

- To overcome the limitations of DPKD algorithm for model compression, we consider a framework for evolving the larger models to compressed models via private adaptation of Iterative Magnitude Pruning (DPIMP). We show that on standard GLUE benchmarks using BERT models, DPIMP framework produces compressed models whose performance is comparable to larger models at 50% unstructured sparsity levels.

- As a byproduct of DPIMP approach for model compression, our work also shows that pre-trained BERT models have sparse subnetworks that can be found via DPSGD that have *almost* matching performances of the private full model, similar to the Lottery Ticket Hypothesis for BERT models in non-private settings [20, 11].

To the best of our knowledge, no prior work have studied model compression techniques of LLMs in private settings. A problem broadly related to model compression is ensemble learning, where the goal is to transfer knowledge from an ensemble of teacher models to a single student model [14]. This problem was studied in the private setting by [48, 49], who proposed the Private Aggregation of Teacher Ensembles (PATE) framework. In PATE an ensemble of teacher models is trained on *disjoint private data* and the student model is trained by noisy aggregation of teachers' answers. Two recent works combine PATE framework with KD algorithm for doing noisy aggregation of teachers' answers for mobile analytics and text generation problems [38, 63]. The PATE framework and ensemble learning techniques can be applied for fine-tuning (or training) deep learning models. Unfortunately, however, as previous works have shown, the performance of deep learning models trained via PATE

are inferior to that of DPSGD for complex datasets [76]. As the performance of a fine-tuned large model on a sensitive dataset is an upper bound on the performance of a compressed model, and we do not know how to fine-tune large models using PATE to match the performance of DPSGD, we do not consider PATE framework for model compression.

## 2  Preliminaries

Recall the formal definition of differential privacy.

**Definition 2.1** (Differential Privacy (DP) [17, 16]). *A randomized algorithm $\mathcal{A}$ is ($\epsilon,\delta$)-differentially private if for any two neighboring datasets $D$ and $D'$, which differ in exactly the data pertaining to a single user, and for all sets $\mathcal{S}$ of possible outputs:* $\Pr[\mathcal{A}(D) \in \mathcal{S}] \leq e^\epsilon \Pr[\mathcal{A}(D') \in \mathcal{S}] + \delta$.

We train all our models via DPSGD as the optimizer. We briefly describe the algorithm.

### 2.1  Training via DPSGD

To train a deep learning model with privacy, the most widely used algorithm is the DP stochastic gradient descent (DPSGD) [55, 4, 1, 52]. DPSGD augments the standard SGD algorithm with per-example gradient clipping and Gaussian noise addition steps. These two steps serve to limit and mask the contribution of a single example. At a high level, the privacy analysis of DPSGD proceeds by first showing that each iteration of DPSGD is differentially private for some $(\epsilon, \delta)$, then applying amplification by subsampling and composition across all the iterations. To get the tightest privacy parameters, however, one needs more sophisticated arguments such as the Moments Accountant method [1] or numerical composition algorithms [23].

## 3  Problem Statement

Input to our problem are privacy parameters $\epsilon > 0$, $\delta > 0$, a large model $M_A$ with the initial model parameters $\theta_A(0)$, a private dataset $D$ corresponding to a downstream task we want to solve, and a compression factor $\gamma$. Let $|M_A|$ denote the parameter count of $M_A$. Our goal is to produce a compressed model $M_B$ satisfying two constraints: (i) $|M_B| \leq \gamma \cdot |M_A|$ and (ii) the final weights of model $M_B$ (denoted by $\theta_B(t)$) should be ($\epsilon, \delta$)-differentially private with respect to dataset $D$. A compression algorithm can make use of $M_A$ in an arbitrary way as long the final weights of model $M_B$ ($\theta_B(t)$) are differentially private with respect to dataset $D$.

We measure the quality of compression algorithms by comparing the accuracy obtained by $M_B$ satisfying ($\epsilon, \delta$)-DP on downstream task $D$ to the accuracy obtained by $M_A$ satisfying ($\epsilon, \delta$)-DP on downstream task $D$. This allows us to quantify how much performance one loses in private training due to model compression. Note that we are not comparing against the performance of non-private models. We would like to find compression algorithms where differentially private $M_B$ has nearly the same performance as differentially private $M_A$.

## 4  Compressed Models via Knowledge Distillation

One of the most widely used algorithms for compressing models is knowledge distillation (KD) [28, 6, 51]. In this section, we propose a framework for implementing knowledge distillation with DP constraints and evaluate its effectiveness on standard GLUE benchmarks. We begin by briefly describing how the KD algorithm is applied for compressing models; we refer the readers to [24, 43] for more details. Adopting the naming convention from the literature, for the rest of the paper, we call large pre-trained models as *teacher models* and compressed smaller models as *student models*.

### 4.1  Non-Private Knowledge Distillation

Let $\mathcal{T}$ be a teacher network with the class probabilities $P_{\mathcal{T}} = \text{softmax}(a_{\mathcal{T}})$ (a.k.a. soft labels) where $a_{\mathcal{T}}$ is the output of last layer before the softmax operation. Similarly, let $\mathcal{S}$ be a student network with parameters $W_{\mathcal{S}}$ and class probabilities $P_{\mathcal{S}} = \text{softmax}(a_{\mathcal{S}})$. The main idea behind KD algorithm is to train $\mathcal{S}$ to mimic the output distribution of the teacher $P_{\mathcal{T}}$ and the true labels. The intuition is that $P_{\mathcal{T}}$

captures the knowledge learnt by the teacher, in particular probabilities assigned by the teacher to labels that are different from the true label. Hinton et al. [28] suggested to use softmax-temperature where probability for class $i$ of the teacher is given by

$$p_i = \frac{exp(z_i/T)}{\sum_j exp(z_j/T)}$$

with logits $z_i$ where $T$ controls the smoothness of the output distribution. Setting a higher value for the temperature parameter $T$ produces a softer probability distribution over classes. The same relaxation is applied to the output of the student network. The student is trained to minimize the weighted combination of the distillation loss and the supervised training loss:

$$\mathcal{L}_{\text{KD}}(\mathcal{W}_\mathcal{S}) := \mathcal{H}(\mathbf{y}_{\text{true}}, P_\mathcal{S}) + \lambda \cdot \mathcal{H}(P_\mathcal{T}, P_\mathcal{S}) \tag{1}$$

where $\mathcal{H}$ refers to the cross-entropy and $\lambda$ is a hyperparameter.

## 4.2 Differentially Private Knowledge Distillation (DPKD)

A natural way to generalize KD algorithm for private distillation of student model is to train the student with DPSGD to minimize the loss function given by Equation 1. However, such an algorithm fails to produce student models satisfying DP because of the term $\mathcal{H}(P_\mathcal{T}, P_\mathcal{S})$ in Equation 1. Note that $P_\mathcal{T}$ is a function of the entire dataset, hence clipping and adding noise alone is not enough to argue that DPSGD produces a private student. A natural solution to overcome this hurdle is to first train the teacher models with DPSGD and then apply KD. We propose our DPKD framework in Algorithm 1. In this Algorithm, if the initialization of student model weights does not incur any privacy cost (e.g. random initialization or initialization using parameters from a publicly trained model), $\epsilon_2$ would be $0$. Having said that, there could be student initialization strategies that are functions of the dataset $D$, in which case, we need to account for the privacy loss using non-zero $\epsilon_2$.

---

**Algorithm 1** Differentially Private Knowledge Distillation (DPKD)

---

**Input:** Teacher model $\mathcal{T}$, student model $\mathcal{S}$, private data $D$, privacy budget $(\epsilon, \delta)$
**Output:** Student model $\mathcal{S}$ satisfying $(\epsilon, \delta)$-DP
 1: Find an allocation of $(\epsilon_1, \delta_1)$, $(\epsilon_2, \delta_2)$ and $(\epsilon_3, \delta_3)$ from the privacy budget $(\epsilon, \delta)$
 2: Train $\mathcal{T}$ on $D$ with DPSGD using privacy budget of $(\epsilon_1, \delta_1)$
 3: Initialize $\mathcal{S}$ (possibly privately with privacy budget $(\epsilon_2, \delta_2)$)
 4: Train $\mathcal{S}$ on $D$ to minimize Eq. (1) with DPSGD using privacy budget of $(\epsilon_3, \delta_3)$
 5: **return** $\mathcal{S}$

---

As the model parameters produced by DPSGD satisfy privacy guarantees, using the post-processing property of DP [18], one can show the following theorem. We omit the proof.

**Theorem 4.1.** *The output of DPKD algorithm is differentially private with privacy parameters obtained by the adaptive composition of privacy parameters in steps 2, 3, and 4 of Algorithm 1.*

In this work we use numerical composition of privacy mechanisms as given in [23]. Our framework for DPKD raises several interesting algorithmic and hyperparameter tuning questions. The most interesting one is whether one can have DPKD algorithm where the privacy budget is not *wasted* on training the teacher. While this is a nice theoretical question, we show in our experiments that DPKD framework is competitive with respect to teachers that are *not trained with DP*.

## 4.3 Empirical Evaluation of DPKD Algorithm

In this section, we perform experiments to evaluate the effectiveness of DPKD algorithm for compressing models.

**Teacher and Student Architectures.** Our teacher models are pre-trained BERT [1] models, which consists of 12 transformer blocks. The architecture of compressed models consist of 6 transformer blocks, which we refer to as $\frac{1}{2}$-BERT.

---

[1]We use Huggingface's `bert-base-uncased`.

Table 1: Comparison between the performance of 6-layer $\frac{1}{2}$-BERT student models with random initialization against full 12-layer BERT teacher models. The first row indicates the performance of fine-tuning the full teacher model. All our models have the same privacy budget $\epsilon = 4$.

| Model | Initialization | Teacher | Training | MNLI | QQP | QNLI | SST-2 | Avg |
|---|---|---|---|---|---|---|---|---|
| BERT | Pretrained | - | Finetune | 77.8 | 84.7 | 87.8 | 90.5 | 85.2 |
| $\frac{1}{2}$-BERT | Random | - | Finetune | 55.1 | 74.0 | 59.4 | 69.7 | 64.5 |
| $\frac{1}{2}$-BERT | Random | BERT | DPKD | 53.9 | 73.1 | 59.2 | 65.4 | 62.9 |

**Tasks and datasets.** Following prior work [77, 34, 75], we experiment with the following set of 4 tasks from the GLUE benchmark [68]: MNLI (Multi-Genre Natural Language Inference Corpus), QQP (Quora Question Pairs), QNLI (Stanford Question Answering Dataset) and SST-2 (Stanford Sentiment Treebank).

**Training and privacy parameters.** We perform experiments with two sets of privacy budgets: (i) $\epsilon = 4$ with $\delta = \frac{1}{N}$ and (ii) $\epsilon = 1$ with $\delta = \frac{1}{10N}$, where $N$ is the number of samples in the given dataset. We describe the software and hardware specifications in Appendix A.1 and hyperparameter settings in Appendix A.2.

Table 2: Comparison between the performance of 6-layer $\frac{1}{2}$-BERT student models with random initialization against full 12-layer BERT teacher models. The first row indicates the performance of fine-tuning the full teacher model. All our models have the same privacy budget $\epsilon = 1$.

| Model | Initialization | Teacher | Training | MNLI | QQP | QNLI | SST-2 | Avg |
|---|---|---|---|---|---|---|---|---|
| BERT | Pretrained | - | Finetune | 74.8 | 82.1 | 85.6 | 86.8 | 82.3 |
| $\frac{1}{2}$-BERT | Random | - | Finetune | 49.6 | 72.6 | 57.8 | 51 | 57.7 |
| $\frac{1}{2}$-BERT | Random | BERT | DPKD | 46.4 | 70.4 | 52.9 | 52 | 55.4 |

We start with *random initialization* of the student models to see if DPKD algorithm can be effective in transferring the knowledge from the teacher. We compare the performance of our student model trained with DPKD algorithm with the performance of directly fine-tuned student via DPSGD and the performance of full teacher model fine-tuned with DPSGD. Our results are summarized in Tables 1 and 2. The main takeaway from this experiment is:

- There is a large gap in the performance of students trained using DPKD algorithm compared to the teacher when students are randomly initialized. In fact, directly fine-tuning the student model using DPSGD achieves better performance compared to DPKD algorithm.

We conclude that DPKD alone is not enough to transfer the knowledge from teacher models to compressed student models.

### 4.4 Better Student Models via Zero-shot Initializations

In our desire to train better-performing student models, we explore different initialization strategies. We note that initialization is also a form of knowledge transfer. Here we consider two natural zero-shot initialization strategies:

- Zero-shot (PT): Here we initialize the student model using the weights of the pre-trained teacher. In particular, we simply initialize the layers of the student model with 6 layers of BERT teacher model. We follow [52] for the choice of layers.
- Zero-shot (FT): Here we initialize the student model using weights of the privately fine-tuned teacher model. As our DPKD requires the teacher to be private as well, one can initialize the student model using the weights of the private teacher without incurring any additional privacy cost.

We compare zero-shot initialization strategies against a fully pre-trained student model given by Huggingface's compressed BERT model DistilBERT [52]. DistilBERT is trained using KD algorithm

Table 3: Comparison between the performance of 6-layer $\frac{1}{2}$-BERT student models against full 12-layer BERT teacher models and pre-trained DistilBERT, under various initialization strategies. For every student initialization method, we compare fine-tuning using DPKD algorithm vs full fine-tuning via DPSGD. All our models have the same privacy budget $\epsilon = 4$.

| Model | Initialization | Teacher | Training | MNLI | QQP | QNLI | SST-2 | Avg |
|---|---|---|---|---|---|---|---|---|
| BERT | Pretrained | - | Finetune | 77.8 | 84.7 | 87.8 | 90.5 | 85.2 |
| $\frac{1}{2}$-BERT | Zero-shot (PT) | - | Finetune | 71.7 | 82.4 | 83.2 | 82.7 | 80.0 |
| $\frac{1}{2}$-BERT | Zero-shot (PT) | BERT | DPKD | 72.8 | 82.6 | 83.0 | 82.7 | 80.3 |
| $\frac{1}{2}$-BERT | Zero-shot (FT) | - | Finetune | 71.3 | 81.8 | 83.4 | 82.2 | 79.7 |
| $\frac{1}{2}$-BERT | Zero-shot (FT) | BERT | DPKD | 72.3 | 82.1 | 82.9 | 82.6 | 80.0 |
| DistilBERT | Pretrained | - | Finetune | 73.0 | 84.3 | 82.8 | 87.7 | 81.9 |
| DistilBERT | Pretrained | BERT | DPKD | 72.9 | 83.7 | 83.0 | 86.6 | 81.5 |

Table 4: Comparison between the performance of 6-layer $\frac{1}{2}$-BERT student models against full 12-layer BERT teacher models and pre-trained DistilBERT, under various initialization strategies. For every student initialization method, we compare fine-tuning using DPKD algorithm vs full fine-tuning via DPSGD. All our models have the same privacy budget $\epsilon = 1$.

| Model | Initialization | Teacher | Training | MNLI | QQP | QNLI | SST-2 | Avg |
|---|---|---|---|---|---|---|---|---|
| BERT | Pretrained | - | Finetune | 74.8 | 82.1 | 85.6 | 86.8 | 82.3 |
| $\frac{1}{2}$-BERT | Zero-shot (PT) | - | Finetune | 66.9 | 78.3 | 81.0 | 79.6 | 76.4 |
| $\frac{1}{2}$-BERT | Zero-shot (PT) | BERT | DPKD | 67.5 | 78.4 | 80.1 | 78.5 | 76.1 |
| $\frac{1}{2}$-BERT | Zero-shot (FT) | - | Finetune | 66.9 | 77.6 | 80.1 | 79.2 | 75.9 |
| $\frac{1}{2}$-BERT | Zero-shot (FT) | BERT | DPKD | 68.3 | 77.0 | 80.3 | 80.0 | 76.4 |
| DistilBERT | Pretrained | - | Finetune | 68.4 | 82.0 | 81.0 | 86.0 | 79.3 |
| DistilBERT | Pretrained | BERT | DPKD | 68.1 | 80.5 | 80.2 | 85.1 | 78.5 |

with the full BERT model on public data at the *pre-training stage*. We note that DistilBERT has 6 transformer blocks and hence has the same architecture as $\frac{1}{2}$-BERT. We emphasize that pre-training student model is computationally expensive, and the main aim of this paper is to develop algorithms where one can transfer knowledge from the teacher to students models without resorting to pre-training the student from scratch. However, DistilBERT serves as the gold standard to compare our zero-shot initialization strategies and also sets the benchmark for comparing ideas in Section 5.

**Results** Our results are summarized in Tables 3 and 4. For every student initialization method, we also compare fine-tuning using DPKD algorithm vs simply fine-tuning the student via DPSGD. The main takeaways from these experiments are:

- Zero-shot initialization strategies give large performance improvements to student models and come to 2-3% of the performance achieved by pre-trained DistilBERT. Somewhat surprisingly, there is not much difference between our two zero-shot initialization strategies.

- Both pre-trained DistilBERT and student models with zero-shot initialization strategies fall short of matching the performance of teacher models.

- Finally, broadly speaking, DPKD algorithm does not give a significant performance boost to the student models compared to directly fine-tuning them.

### 4.5 Better Models are Better Teachers?

Given the results in the previous section, one may wonder if better teachers can help in improving the performance of the students. In this regard, we consider two notions of *better teacher*: (1) A

Table 5: Comparison between the performance of 6-layer $\frac{1}{2}$-BERT student models under different teacher models in distillation against full 12-layer BERT teacher models and pre-trained DistillBERT. All our models have the same privacy budget $\epsilon = 4$.

| Model | Initialization | Teacher | Training | MNLI | QQP | QNLI | SST-2 | Avg |
|---|---|---|---|---|---|---|---|---|
| BERT | Pretrained | - | Finetune | 77.8 | 84.7 | 87.8 | 90.5 | 85.2 |
| Student | Pretrained | - | Finetune | 73.0 | 84.3 | 82.8 | 87.7 | 81.9 |
| $\frac{1}{2}$-BERT | Zero-shot (PT) | BERT | DPKD | 72.8 | 82.6 | 83.0 | 82.7 | 80.3 |
| $\frac{1}{2}$-BERT | Zero-shot (PT) | BERT$_{\text{LARGE}}$ | DPKD | 72.4 | 81.1 | 83.1 | 81.5 | 79.5 |
| $\frac{1}{2}$-BERT | Zero-shot (PT) | BERT w/out DP | DPKD | 74.2 | 82.9 | 84.5 | 83.0 | 81.1 |

Table 6: Comparison between the performance of 6-layer $\frac{1}{2}$-BERT student models under different teacher models in distillation against full 12-layer BERT teacher models and pre-trained DistillBERT. All our models have the same privacy budget $\epsilon = 1$.

| Model | Initialization | Teacher | Training | MNLI | QQP | QNLI | SST-2 | Avg |
|---|---|---|---|---|---|---|---|---|
| BERT | Pretrained | - | Finetune | 74.8 | 82.1 | 85.6 | 86.8 | 82.3 |
| Student | Pretrained | - | Finetune | 68.4 | 82.0 | 81.0 | 86.0 | 79.3 |
| $\frac{1}{2}$-BERT | Zero-shot (PT) | BERT | DPKD | 67.5 | 78.4 | 80.1 | 78.5 | 76.1 |
| $\frac{1}{2}$-BERT | Zero-shot (PT) | BERT$_{\text{LARGE}}$ | DPKD | 67.6 | 78.0 | 80.1 | 78.0 | 75.9 |
| $\frac{1}{2}$-BERT | Zero-shot (PT) | BERT w/out DP | DPKD | 70.6 | 79.0 | 81.5 | 79.8 | 77.7 |

larger teacher model in Step 2 of Algorithm 1 (2) A teacher model that is not DP-trained. We note that the final student model in the second case is not DP; however, these experiments allow us to quantify how much performance gains one could have if one were to come up with a new framework for implementing KD in DP setting without requiring the teacher to be trained with DP.

Our results are summarized in Tables 5 and 6. These experiments show that DPKD does not benefit significantly from having better teacher models. Furthermore, our proposed framework of implementing KD in DP framework by training both the teacher and student models via DP is only within 0.8% of the doing KD with fine-tuned teacher without DP.

## 5 Evolving Teacher to Student Models via Pruning

As the previous section presents, the Knowledge Distillation approach to model compression has two main drawbacks in the private world:

- Drop in accuracy: There is a considerable drop in the accuracy between the teacher and the student models.

- Good initialization of students is crucial: The best performance is obtained by students who already have a good initialization; in our experiments, pre-trained DistilBERT mostly achieved the best student performance.

Finding a good initialization can be challenging in practice. Often, the student architectures are chosen to suit the hardware and latency requirements of the application for which the model is being deployed, using neural architecture search [19]. Hence, finding a good initialization for every student architecture via pre-training can be expensive and in most cases impossible. Our zero-shot initialization strategies alleviate this problem to a certain degree, yet fall short of closing the gap between the teacher and the student performances. Moreover, DPKD requires that (i) the teacher is trained with DPSGD and (ii) the student is distilled via DPSGD. This two-step approach creates additional overheads in terms of training. Given these limitations, it is natural to ask: Can we evolve the teacher to a student model while fine-tuning with DPSGD? In this section, we explore an answer to this via *structured and unstructured pruning* with privacy, which allows us to obtain student models that are as good as the teacher models.

## 5.1 Model Compression via Pruning

Pruning algorithms are a broad class of model compression techniques where one drops the parameters from a model during or after the training process. Many works have shown that eliminating unnecessary parameters of neural networks via pruning can lead to sparser and compressed models that have shorter inference times without loss in performance [32, 27, 26]. For example, in *magnitude pruning*, one of the most widely used pruning techniques, we prune a fraction of parameters with the lowest magnitude. However, there are several pruning strategies, and we refer the readers to [35] for more details and references.

Pruning can be implemented in both *structured and unstructured* ways. In structured pruning, all the pruned weights belong to a single building block of the model. For example, a 6-layer $\frac{1}{2}$-BERT can be obtained by pruning 6 layers from the full BERT model, which consists of 12 transformer blocks. On the other hand, in unstructured pruning, pruned weights may be spread across all the layers of the network. In unstructured pruning, it is possible to obtain a $50\%$ sparse student model while still having all the 12 layers of BERT. Depending on the hardware architectures, inference latency between models with structured and unstructured sparsity could be quite different. However, in this section, *we use sparsity as the main measure of model compression*, which is also well accepted in the community [35, 29].

## 5.2 Iterative Magnitude Pruning (IMP)

Private pruning techniques we study in this section are based on the Iterative Magnitude Pruning (IMP) method, which is a specific pruning technique proposed in a recent work on Lottery Ticket Hypothesis [20]. The idea behind IMP is rather simple: As we train a deep learning model, after every $N$ iterations we prune an $\alpha\%$ of the weights with the *lowest magnitude*. We repeat this process until we achieve the desired sparsity. Here, both $N$ and $\alpha$ are hyperparameters that need to be tuned. For example, to achieve $50\%$ sparsity, one can perform $5N$ iterations where after every $N$ iterations additional $10\%$ of the weights with the least magnitudes are dropped. As specified IMP produces unstructured sparsity. However, we consider a simple modification of the IMP algorithm to produce structured sparsity as well.

---

**Algorithm 2** Structured DPIMP

**Input:** Teacher model $\mathcal{T}$, number of layers to prune $L$, hyperparams $\alpha$, $N$ and $M$
**Output:** Private student model $\mathcal{S}$ with $L$ layers pruned from $\mathcal{T}$
 1: Set $\mathcal{S} := \mathcal{T}$
 2: **for** $j = 1$ to $L$ **do**
 3:     Fine-tune $\mathcal{S}$ for $N$ iterations with DPSGD
 4:     Set $W_{\min}$ consisting of $\alpha\%$ of the remaining model weights with the least magnitude
 5:     Set $W_i$ as the weights of layer $i$
 6:     Drop the layer $i^*$ from $\mathcal{S}$ satisfying $i^* := \arg\max_i \{W_i \cap W_{\min}\}$
 7: **end for**
 8: Fine-tune $\mathcal{S}$ for $M$ more iterations with DPSGD
 9: **return** $\mathcal{S}$

---

## 5.3 Structured DPIMP

We first attempt to obtain a student model from the teacher model via a structured IMP technique, using the following modification: During fine-tuning the teacher model with DPSGD, we progressively drop an *appropriately chosen transformer block* from the teacher model at the end of every $N$ iterations. We repeat this process until we obtain the student model with the required sparsity. The layer to drop is chosen using the following heuristic: Let $\alpha > 0$ be a hyperparameter. At the end of $N$ iterations, fix bottom (by magnitude) $\alpha\%$ of *all* model weights, and denote it by $W_{\min}$. For the $i^{th}$ transformer block, let $W_i$ denote the set of model weights belonging to that block. Among all the transformers blocks we find the block $i^*$ that has the highest number of weights from the set $W_{\min}$; Formally, $i^* := \arg\max_i \{W_i \cap W_{\min}\}$, and we prune the transformer layer $i^*$. We present this in Algorithm 2. We note that the algorithm satisfies $(\epsilon, \delta)$-DP after the pruning steps due to the post-processing property of differential privacy.

Table 7: Comparing performance of 6-layer $\frac{1}{2}$-BERT student model produced by structured DPIMP with 12-layer BERT teacher model and pre-trained DistillBERT. The results are shown with two privacy budgets $\epsilon = 1$ and $\epsilon = 4$.

| Model | MNLI | | QQP | | QNLI | | SST-2 | | Avg | |
|---|---|---|---|---|---|---|---|---|---|---|
| | $\epsilon = 1$ | $\epsilon = 4$ | $\epsilon = 1$ | $\epsilon = 4$ | $\epsilon = 1$ | $\epsilon = 4$ | $\epsilon = 1$ | $\epsilon = 4$ | $\epsilon = 1$ | $\epsilon = 4$ |
| BERT | 74.8 | 77.8 | 82.1 | 84.7 | 85.6 | 87.8 | 86.8 | 90.5 | 82.3 | 85.2 |
| DistilBERT | 68.4 | 73.0 | 82.0 | 84.3 | 81.0 | 82.8 | 86.0 | 87.7 | 79.3 | 81.9 |
| $\frac{1}{2}$-BERT | 68.7 | 72.9 | 80.7 | 83.1 | 80.9 | 82.5 | 83.3 | 85.7 | 78.4 | 81.0 |

**Empirical Evaluation**    We evaluate our structured pruning algorithm with the same setup described in Section 4.3, and provide the hyperparameters in Appendix A.2. We split the privacy budget equally among all the iterations of the algorithm. Our goal is to produce a student model which has $\frac{1}{2}$ as many layers as the full BERT model. Table 7 shows the results for this setting where we compare structured DPIMP to private fine-tuning of the pre-trained DistilBERT and the full BERT model. The main takeaway from this experiment is:

- DP structured pruning algorithm produces a student model that has performance comparable to that of DistilBERT. Further, it avoids the pre-training cost associated with DistilBERT.

## 5.4    Unstructured DPIMP

The structured pruning produces student models that are as good as DistilBERT; our next target is to produce student models that are as good as the full teacher BERT model. Towards that, we explore unstructured pruning techniques, in particular differentially private version of the IMP algorithm. As we fine-tune the deep learning models using DPSGD, after every $N$ iterations we prune increments of $\alpha\%$ of the weights with lowest magnitude. The remaining weights are then reset to the original pre-trained initialization. (We do this step to establish connections to Lottery Ticket Hypothesis, see below.) We repeat this process until we achieve the desired sparsity. We note that the initialization of the final student model's non-zero parameters at the end of the pruning step are still non-private, even though weights correspond to pre-trained model weights. This is due to the fact that pruned parameters are found during the fine-tuning process, which makes use of the private dataset. Therefore, the whole process must be performed with DPSGD to produce a private student model. Finally, we perform additional fine-tuning of the pruned student model by using DPSGD for $M$ more iterations. We call the private variation of this procedure Unstructured DPIMP. We formally present this process in Algorithm 3.

---

**Algorithm 3** Unstructured DPIMP

**Input:**  Model $\mathcal{T}$ and the sparsity level $S\%$, hyperparams $\alpha$, $N$ and $M$
**Output:**  Private Student model $\mathcal{S}$ satisfying the sparsity requirements
 1: Set $\mathcal{T}' := \mathcal{T}$
 2: **for** $i = 1$ to $\lceil (S/\alpha) \rceil$ **do**
 3:     Fine-tune $\mathcal{T}'$ for $N$ iterations with DPSGD
 4:     Prune $(\alpha \times i)\%$ of weights with the lowest magnitude from $\mathcal{T}'$
 5:     Reset the non-zero weights of $\mathcal{T}'$ to the original $\mathcal{T}$
 6: **end for**
 7: Fine-tune $\mathcal{T}'$ for $M$ iterations with DPSGD
 8: **return** $\mathcal{S} := \mathcal{T}'$

---

**Empirical Evaluation**    We refer to the student model produced by Algorithm 3 as SparseBERT and indicate the sparsity level in brackets. We allocate the privacy budget equally among all iterations of Algorithm 3. Table 8 summarizes our experiments on unstructured pruning via DPIMP. We compare the performance of our student model to pre-trained DistilBERT (whose parameter count is the same as our final student model) and the full-sized BERT teacher model. The main takeaways are:

- DPIMP produces a student model that has better performance compared to DistilBERT.

Table 8: Comparing performance of SparseBERT student model produced by unstructured DPIMP with 12-layer BERT teacher and pre-trained DistillBERT. The results are shown with two privacy budgets $\epsilon = 1$ and $\epsilon = 4$.

| Model | MNLI | | QQP | | QNLI | | SST-2 | | Avg | |
|---|---|---|---|---|---|---|---|---|---|---|
| | $\epsilon = 1$ | $\epsilon = 4$ | $\epsilon = 1$ | $\epsilon = 4$ | $\epsilon = 1$ | $\epsilon = 4$ | $\epsilon = 1$ | $\epsilon = 4$ | $\epsilon = 1$ | $\epsilon = 4$ |
| BERT | 74.8 | 77.8 | 82.1 | 84.7 | 85.6 | 87.8 | 86.8 | 90.5 | 82.3 | 85.2 |
| DistilBERT | 68.4 | 73.0 | 82.0 | 84.3 | 81.0 | 82.8 | 86.0 | 87.7 | 79.3 | 81.9 |
| SparseBERT (50%) | 72.9 | 76.8 | 81.1 | 84.0 | 82.2 | 85.7 | 83.7 | 87.6 | 80.0 | 83.5 |

- The average performance of DPIMP is within $2\%$ of the full BERT model. We conclude that unstructured pruning techniques are more effective in closing the gap between the teacher and the student models in the private world. Moreover, pruning methods are computationally cheaper as student models do not require pre-training on public data.

- DPIMP algorithm as described in Algorithm 3 at the end of line 6 finds sparse subnetworks in the pre-trained BERT model that can be fine-tuned to obtain nearly matching teacher performance. This is similar to Lottery Ticket Hypothesis work for BERT networks [11], although in the private world performance of the student network is not matching that of the teacher.

## 6 Conclusions and Future Directions

In this paper we initiated the study of differentially private model compression via Knowledge Distillation and Pruning, and gave frameworks for implementing both. Our work shows that one can obtain student models whose performance comes within $2\%$ of the full teacher models while having $50\%$ fewer parameters, which can lead to significant reduction in memory and improve inference time. We believe that our work takes the first step in the intersection of private training and model compression techniques, and opens a whole in direction with plenty of interesting and important problems. We highlight some of them here.

- *Better Algorithms For Model Compression*: While we gave natural DP adaptations of KD and pruning algorithms, we believe that there are plenty of new techniques to explore.

- *Better Accounting*: In all our results, we chose very simple strategies for allocating privacy budget across various steps of training and pruning. Theoretical innovations in the form of better accounting can further improve the performance of student models.

- *Lottery Tickets for DP-training?* Both in our experiments on KD and in pruning, we see that models that have "good initialization" lead to dramatic improvements in performance. Moreover our DPIMP algorithm finds sparse subnetworks in pretrained BERT models at $50\%$ sparsity that can be fine-tuned to obtain *nearly* matching teacher performances. This raises an intriguing question akin to Lottery Ticket Hypothesis: Are there good initialization of models where dynamics of DPSGD is similar to SGD? We believe that this is an exciting research direction both from a theoretical point of view and also its implications to practical private training.

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
