Table 9: Hyper-parameters for all MNLI Experiments.

| Model | Initialization | Teacher | Training | Epochs | LR | NM |
|---|---|---|---|---|---|---|
| BERT$_{\text{BASE}}$ | - | - | Finetune | 50 | 0.0001 | 0.7811 |
| $\frac{1}{2}$-BERT | Random | - | Finetune | 75 | 0.0001 | 0.841 |
| $\frac{1}{2}$-BERT | Random | BERT$_{\text{BASE}}$ | DPKD | 25+50 | 0.0001 | 0.841 |
| $\frac{1}{2}$-BERT | Zero-shot (PT) | - | Finetune | 75 | 0.0001 | 0.841 |
| $\frac{1}{2}$-BERT | Zero-shot (PT) | BERT$_{\text{BASE}}$ | DPKD | 25+50 | 0.0001 | 0.841 |
| $\frac{1}{2}$-BERT | Zero-shot (FT) | - | Finetune | 25+50 | 0.0001 | 0.841 |
| $\frac{1}{2}$-BERT | Zero-shot (FT) | BERT$_{\text{BASE}}$ | DPKD | 25+50 | 0.0001 | 0.841 |
| DistilBERT | Pretrained | - | Finetune | 75 | 0.0001 | 0.841 |
| DistilBERT | Pretrained | BERT$_{\text{BASE}}$ | DPKD | 25+50 | 0.0001 | 0.841 |
| $\frac{1}{2}$-BERT | Zero-shot (PT) | BERT$_{\text{LARGE}}$ | DPKD | 25+50 | 0.0001 | 0.841 |
| $\frac{1}{2}$-BERT | Zero-shot (PT) | BERT$_{\text{BASE}}$ without DP | DPKD | 10+50 | 0.0001 | 0.803 |
| $\frac{1}{2}$-BERT | - | - | Structured DPIMP | 25+25 | 0.00008 | 0.781 |
| SparseBERT | - | - | Unstructured DPIMP | 15+50 | 0.0001 | 0.815 |

# A   Appendix

## A.1   Software and Hardware Specifications

We use Opacus 0.15.0, Huggingface Transformers 4.10.3, PyTorch 1.9.1 with Cuda 10.2, and Python 3.8.8. We run our experiments using PyTorch's distributed training on an Azure ML Nvidia DGX-2 system, which has 16 Tesla V100 GPUs with 512GB memory in total.

## A.2   Hyper-parameters

In this section we present all the hyper-parameters used for training our models. Tables 9, 10, 11 and 12 show hyper-parameters used to produce the results in Tables 3, 5, 7, and 8. As a general note, our experimental framework entails a large combinatorial search space for the hyper-parameters, therefore, we take into account of the findings of prior work to be more efficient in this regard.

We fix the gradient norm to be 1 and set the batch size as 1024 in all experiments based on [75, 34].

In distillation experiments, we observed that longer training (especially for distillation part) helps improving the performance. Therefore, we set the total number of epochs to 75 (and 40 for SST-2 due to its smaller size in comparison). We compute the corresponding noise multiplier (NM) so that the privacy budget gives $\epsilon = 4$ with $\delta = \frac{1}{N}$ where $N$ is the number of samples in the given dataset. We state the epochs in the distillation experiments as $x + y$ format, where $x$ corresponds to the training epochs for the teacher and $y$ corresponds to distillation into the student. We simply set $x$ to be one third of the total number of epochs. We did not spend any hyper-parameter search on this part as the performance with this setting provided close performance to the case when the teacher is trained without DP (hence all privacy budget is spent on distillation), which is an upper bound to private distillation framework.

For pruning experiments, the $x + y$ epoch format shows the epochs spent on pruning, $x$ and on fine-tuning, $y$. $x$ epochs are divided equally in the for loop of Algorithm 2 and 3. $y$ epochs correspond to $M$ iterations in Algorithm 2 and 3. We set $x = 15$ and $\alpha = 10$ based on [11].

Finally, learning rate is an important parameter to be tuned [75]. Hence, we ran a grid search over the rates $0.0005, 0.0008, 0.001, 0.002$ and picked the best one.

# B   Extended Related Work

**Model Compression.**   Work on model compression can be roughly divided into three main categories: distillation, pruning, and quantization, where quantization is orthogonal to the first two and can be applied on top of them as well [21]. Closely related to compression but outside the scope of

Table 10: Hyper-parameters for QQP

| Model | Initialization | Teacher | Training | Epochs | LR | NM |
|---|---|---|---|---|---|---|
| BERT$_{BASE}$ | - | - | Finetune | 75 | 0.0001 | 0.856 |
| $\frac{1}{2}$-BERT | Random | - | Finetune | 75 | 0.0001 | 0.856 |
| $\frac{1}{2}$-BERT | Random | BERT$_{BASE}$ | DPKD | 25+50 | 0.0001 | 0.856 |
| $\frac{1}{2}$-BERT | Zero-shot (PT) | - | Finetune | 75 | 0.0001 | 0.856 |
| $\frac{1}{2}$-BERT | Zero-shot (PT) | BERT$_{BASE}$ | DPKD | 25+50 | 0.0001 | 0.856 |
| $\frac{1}{2}$-BERT | Zero-shot (FT) | - | Finetune | 25+50 | 0.0001 | 0.856 |
| $\frac{1}{2}$-BERT | Zero-shot (FT) | BERT$_{BASE}$ | DPKD | 25+50 | 0.0001 | 0.856 |
| DistilBERT | Pretrained | - | Finetune | 75 | 0.0001 | 0.856 |
| DistilBERT | Pretrained | BERT$_{BASE}$ | DPKD | 25+50 | 0.0001 | 0.856 |
| $\frac{1}{2}$-BERT | Zero-shot (PT) | BERT$_{LARGE}$ | DPKD | 25+50 | 0.0001 | 0.856 |
| $\frac{1}{2}$-BERT | Zero-shot (PT) | BERT$_{BASE}$ without DP | DPKD | 10+50 | 0.0001 | 0.815 |
| $\frac{1}{2}$-BERT | - | - | Structured DPIMP | 25+25 | 0.00008 | 0.790 |
| SparseBERT | - | - | Unstructured DPIMP | 15+50 | 0.0001 | 0.829 |

Table 11: Hyper-parameters for QNLI

| Model | Initialization | Teacher | Training | Epochs | LR | NM |
|---|---|---|---|---|---|---|
| BERT$_{BASE}$ | - | - | Finetune | 75 | 0.0001 | 1.2 |
| $\frac{1}{2}$-BERT | Random | - | Finetune | 75 | 0.0001 | 1.2 |
| $\frac{1}{2}$-BERT | Random | BERT$_{BASE}$ | DPKD | 25+50 | 0.0001 | 1.2 |
| $\frac{1}{2}$-BERT | Zero-shot (PT) | - | Finetune | 75 | 0.0001 | 1.2 |
| $\frac{1}{2}$-BERT | Zero-shot (PT) | BERT$_{BASE}$ | DPKD | 25+50 | 0.0001 | 1.2 |
| $\frac{1}{2}$-BERT | Zero-shot (FT) | - | Finetune | 25+50 | 0.0001 | 1.2 |
| $\frac{1}{2}$-BERT | Zero-shot (FT) | BERT$_{BASE}$ | DPKD | 25+50 | 0.0001 | 1.2 |
| DistilBERT | Pretrained | - | Finetune | 75 | 0.0001 | 1.2 |
| DistilBERT | Pretrained | BERT$_{BASE}$ | DPKD | 25+50 | 0.0001 | 1.2 |
| $\frac{1}{2}$-BERT | Zero-shot (PT) | BERT$_{LARGE}$ | DPKD | 25+50 | 0.0001 | 1.2 |
| $\frac{1}{2}$-BERT | Zero-shot (PT) | BERT$_{BASE}$ without DP | DPKD | 10+50 | 0.0001 | 1.074 |
| $\frac{1}{2}$-BERT | - | - | Structured DPIMP | 25+25 | 0.00008 | 1.074 |
| SparseBERT | - | - | Unstructured DPIMP | 15+50 | 0.0001 | 1.162 |

this paper is neural architecture search [72, 19, 74, 61], which attempts to automate the process of designing new neural architectures, with high performance and low computation/memory costs.

Distillation: knowledge distillation is most commonly used on output logits to train smaller BERT models using the logits of a larger, higher accuracy teacher [58, 52, 31, 65, 7, 73, 54, 40, 71, 15, 47]. Knowledge distillation is also used for training BiLSTM models, as a faster alternative to Transformers [70]. Compressing knowledge to a BiLSTM is typically done directly for a specific NLP task [46]. Since BiLSTMS are usually trained from scratch on different tasks, several different techniques are proposed to generate additional synthetic training data. [62, 45] use rule-based data augmentation while [36] user data collected from multiple tasks to train a single model.

Pruning: Pruning [29, 11] is a technique that discovers, and then eliminates, redundant or unimportant weights, layers, or other components. Pruning not only improves the prediction time of the model, but it also sometimes makes the model more robust and more performant [21]. Prior work that study pruning in the context of BERT [11] can be categorized into unstructured or structured pruning methods. Those that prune individual weights are unstructured while structured methods prune structured blocks of weights [33] or even complete layers in the BERT model. Structured pruning can be done by pruning attention heads, pruning encoder units, or pruning the embedding layer.

Table 12: Hyper-parameters for SST-2

| Model | Initialization | Teacher | Training | Epochs | LR | NM |
|---|---|---|---|---|---|---|
| BERT$_{\text{BASE}}$ | - | - | Finetune | 40 | 0.0001 | 1.13 |
| $\frac{1}{2}$-BERT | Random | - | Finetune | 40 | 0.0001 | 1.13 |
| $\frac{1}{2}$-BERT | Random | BERT$_{\text{BASE}}$ | DPKD | 15+25 | 0.0001 | 1.13 |
| $\frac{1}{2}$-BERT | Zero-shot (PT) | - | Finetune | 40 | 0.0001 | 1.13 |
| $\frac{1}{2}$-BERT | Zero-shot (PT) | BERT$_{\text{BASE}}$ | DPKD | 15+25 | 0.0001 | 1.13 |
| $\frac{1}{2}$-BERT | Zero-shot (FT) | - | Finetune | 15+25 | 0.0001 | 1.13 |
| $\frac{1}{2}$-BERT | Zero-shot (FT) | BERT$_{\text{BASE}}$ | DPKD | 15+25 | 0.0001 | 1.13 |
| DistilBERT | Pretrained | - | Finetune | 40 | 0.0001 | 1.13 |
| DistilBERT | Pretrained | BERT$_{\text{BASE}}$ | DPKD | 15+25 | 0.0001 | 1.13 |
| $\frac{1}{2}$-BERT | Zero-shot (PT) | BERT$_{\text{LARGE}}$ | DPKD | 15+25 | 0.0001 | 1.13 |
| $\frac{1}{2}$-BERT | Zero-shot (PT) | BERT$_{\text{BASE}}$ without DP | DPKD | 10+25 | 0.0001 | 1 |
| $\frac{1}{2}$-BERT | - | - | Structured DPIMP | 16+24 | 0.0001 | 1.13 |
| SparseBERT | - | - | Unstructured DPIMP | 15+25 | 0.0001 | 1.13 |

Unstructured pruning methods include magnitude pruning [11], which simply removes weights close to zero, movement-based pruning [59], which removes weights moving towards zero during fine-tuning, and reweighted proximal pruning (RPP) [25], which uses iteratively reweighted l1 minimization followed by the proximal algorithm for decoupling pruning and error back-propagation.

Quantization: Quantization involves reducing the number of unique values required to represent model weights and activations enabling their representation with only few bits, reducing the memory footprint, and lowering the precision of the numerical calculations. A naive approach to quantization is to simply truncate each weight to the target bitwidth, which often results in a significant drop in accuracy of the model [56]. To mitigate this Quantization-Aware Training (QAT) is used, which involves additional training steps to adjust the quantized weights [78, 60].

**Differentially Private Model Traning.**   Prior related work studies differentially private training [1] of language models from scratch on LSTMs [41, 8], or transformer-based large language models, i.e. LLMs [2, 30]. A more recent line of work studies private fine-tuning of pre-trained LLMs using DPSGD [75, 34]. These works demonstrate that larger pre-trained models have better performance when fine-tuned privately, than smaller models with fewer parameters, which is contrary to what was observed before when training models from scratch [4, 10, 64]. Different from our problem setting is an interesting problem of protecting privacy during prediction time, which is studied in recent work [22, 39]. [69] study model compression with privacy using KD algorithm for image classification problems. However, their setting is different from ours as they assume the availability of public datasets that have similar distribution as the private datasets. Thus, their framework for doing KD is quite different from ours. [57] study model compression in the federated learning framework. Finally, besides privacy, differential privacy algorithms DPSGD and PATE have also been investigated from the fairness perspective by recent work [3, 66].