# OpenReview forum: "Differentially Private Model Compression"
_NeurIPS.cc/2022/Conference — NeurIPS 2022 Accept_

### Official Review · Reviewer_pc64 · 2022-06-21

**Rating:** 5
**Confidence:** 4
**Soundness:** 2 fair
**Presentation:** 3 good
**Contribution:** 2 fair

**Summary:**

This paper explores the links between differential privacy and model compression. Authors propose a framework for training NLP models with high utility by leveraging DP-SGD and knowledge distillation.
This is an experimental paper.

**Questions:**

A number of questions arose during my evaluation of this work. I would like the authors to either point me to the section where I can find the necessary details or to provide an answer outside of the manuscript space:
- What dataset does a teacher control? There was a discussion about PATE needing a disjoint dataset, which may be unrealistic, but I did not find any mention of what are the data assumptions on the teacher model in this work? Moreover, could the authors elaborate why is a disjoint setup considered unrealistic in this setting?
- I am not entirely sure I understand section 2.1: what is the meaning of the 'sophisticate argument(s)'? It strikes me as if the authors included a list of words associated with DP without really explaining them or even linking them together.
- It was unclear to me why in section 3.2 DP-SGD was insufficient? I suspect this links to my qn on data assumptions for the teacher mode and I am unsure whether a teacher model is required to be private or not, so would you kindly clarify?
- In the experiments an epsilon of < 4.25 was chosen: what is the motivation behind it? Why this specific value and how does it relate to prior literature? If I overlooked it, then please point me to a section/reference.

**Limitations:**

In addition to the limitations outlined by authors, my comments are largely explained above and include: Similar prior work, constraints on the data of the teacher model etc.
While overall, the work shows some interesting results and is relatively easy to read, I cannot recommend acceptance until points above are addressed by the authors (particularly wrt novelty and comparison with prior works).

**Strengths And Weaknesses:**

The work is rather easy to follow, with clear motivations of why this technique should be explored, how the scientific community can benefit from private compressed models etc. The method that authors propose seems sound and can prove useful for a broad scientific community. The way the manuscript is structured really aids its interpretability: the motivation is followed by a naive implementation, followed by the comments on why such method does not perform well, followed by another iteration with limitations addressed etc. This in my view makes the paper much friendlier to larger audience.
Authors provide a number of evaluations of their technique in comparison to its non-private counterpart. What I find really commendable is the fact that authors report the results which were not entirely positive and demonstrated their understanding of the limitations of their method. The overall results of compressed models being just slightly behind the non-private counterparts look rather promising.

However, this work contains a number of shortcomings that I think that authors need to address.
Firstly, authors reject the idea of using PATE in their evaluation as DP-SGD has better performance. I am not entirely sure A) why is this a valid metric to reject the work that is the closest DP student-teacher implementation to this manuscript and B) I am not even sure that this statement is necessarily true [4,5]. It seems to me that PATE was avoided by the authors without a strong reason behind it, making me question some parts of the evaluation of the proposed method.
Secondly, I am not convinced that a combination of DP and KD is a particularly novel method. In addition to the aforementioned PATE, I also found a number of works such as [1,2,3] (in fact [1] even shares the name of the framework with this study), which seem to have addressed this issue before in detail (some of which were peer-reviewed previously). So what exactly is the novel contribution of this work which was not previously proposed? Because on a high level, while the compression seems like a fairly useful trait (and a promising result), combining DP-SGD with KD is fairly trivial and has been done before.
One other comment I have is the argument on the 'better models': for DP training a 'bigger' model does not necessarily outperform a smaller one, so I am not sure such argument holds [6].
Minor: inconsistencies with DP-SGD (the original version) and DPSGD.

[1] Lyu, Lingjuan, and Chi-Hua Chen. "Differentially private knowledge distillation for mobile analytics." Proceedings of the 43rd International ACM SIGIR Conference on Research and Development in Information Retrieval. 2020.
[2] Sun, Lichao, and Lingjuan Lyu. "Federated model distillation with noise-free differential privacy." arXiv preprint arXiv:2009.05537 (2020).
[3 ]Wang, Ji, et al. "Private model compression via knowledge distillation." Proceedings of the AAAI Conference on Artificial Intelligence. Vol. 33. No. 01. 2019.
[4] Papernot, Nicolas, et al. "Scalable private learning with pate." arXiv preprint arXiv:1802.08908 (2018).
[5] Uniyal, Archit, et al. "DP-SGD vs PATE: Which Has Less Disparate Impact on Model Accuracy?." arXiv preprint arXiv:2106.12576 (2021).
[6] Klause, Helena, et al. "Differentially private training of residual networks with scale normalisation." arXiv preprint arXiv:2203.00324 (2022).

---

> ### Author Response · Authors · 2022-08-02
> **Response to Reviewer pc64**
>
> We thank you for a detailed review and questions and many positive comments. Before we will address specific questions you asked, we would like to point out that we have a separate comment for all reviewers describing new experiments we performed, the problem we considered in this paper and its comparison to ensemble learning and PATE. That comment should also clarify some of the other questions you had regarding privacy guarantees of our algorithms, what data set teacher models control, etc.
>
> Here we clarify specific questions you asked.
>
> Q: What dataset the teacher controls?
>
> A: Thanks for the question. This should be clear from our formal problem statement in the comment above but we briefly describe it again here. In our problem, both large language models (LLMs or teacher models) and compressed models (student) are working on the same private data set D, which in our case is GLUE tasks: SST2, MNLI, QNLI, QQP. Our LLM models are pretrained models such as BERT, but we have no access to the pretraining dataset. The differential privacy needs to be guaranteed only by the compressed model on the dataset D. The teacher models can use the private dataset D in arbitrary ways as long as published student models are DP. This is exactly the set up considered in the previous works on DP-NLP [34, 74, 76] but with additional constraints on the size of the models.
>
> Q: I am not entirely sure I understand section 2.1: what is the meaning of the 'sophisticate argument(s)'?
>
> A:  Sorry that what we wrote was not clear. Let us recall the line where we use the phrase sophisticated arguments.
>
> “To get the tightest privacy parameters, however, one needs more sophisticated arguments such as the Moments Accountant method [1] or numerical composition algorithms [21]”.
>
> In the above sentence, the phrase sophisticated arguments refer to the Moments Accountant method and numerical composition algorithms. In lines 91-83, we give a high-level description of analysis of DPSGD based on subsampling and strong composition theorem. However, this does not give the tightest bound on privacy. Moments Accountant method and numerical composition algorithms are mathematical techniques of obtaining the tightest composition theorems. These two are main technical contributions of the respective papers, and describing how those two techniques work is out of scope of this paper.
>
> Q: It was unclear to me why in section 3.2 DP-SGD was insufficient?
>
> A:  Thanks for the question. We assume that you are asking why training only student models with DPSGD is not sufficient. Before we answer this question, let us recall the basics of analysis of DPSGD given in lines 87-91. In DPSGD, we add noise to clipped per-sample gradients and hence every iterate of DPSGD is private. In other words, after each iteration t of DPSGD, model weights W_t can be assumed to be public information. Now, in iteration t+1, gradients of samples are computed with respect to W_t, and hence privacy only depends on gradients belonging to the samples in a single batch. This is crucial for applying amplification by subsampling theorems in privacy analysis.
> Now consider a framework where the teacher models are trained using SGD on the dataset D and the student models are trained with DPSGD while minimizing the Equation 1 on dataset D. Such an algorithm does not output a differentially private compressed student model. This is due to the distillation loss term H(y_true, P_S) in Equation 1. Here, P_S is a function of the entire dataset as the teacher was not trained with DP. Therefore,  gradients of samples are now functions of entire dataset D, which forbids us from applying subsampling theorems in privacy analysis. Our solution to circumvent this was to make P_S DP as well by training the teacher model with DP on dataset D.
>
> We hope this explanation helps. This is a subtle but important aspect of our DPKD algorithm.

---

> > ### Author Response · Authors · 2022-08-02
> > **Response to Reviewer pc64 - part II**
> >
> > Q:Comparison to prior work is not exhaustive.
> >
> > A: Thank you for giving references of the papers related to our work. We cite many of those papers in our submission already and we added the other ones pointed out by the reviewer in the revised version. As we wrote in our submission, the settings considered in those references are different from the problem we are considering. Moreover, we are confident that there is no published paper that uses the PATE framework to do fine-tuning of LLMs on GLUE benchmarks and obtains results comparable. Please refer to the response to all reviewers for a more detailed discussion on this point.
> >
> > Q: Choice of privacy budget epsilon.
> >
> > A: Most papers in deep learning space have used epsilon values around [4-8]; see the SOTA papers in NLP and image recognition tasks published last year [34, 74, 12, 42]. Most of those papers use epsilon = 8. Moreover, the US census bureau, which is the largest deployment of DP used epsilon > 18 [X1]. While these epsilon values may seem large, it is still a worst case guarantee. In comparison to many of these works, our choice of epsilon is on the lower end. Moreover, we have conducted more experiments with epsilon = 1, and our results remain unchanged qualitatively; please see Appendix B in the revised version.
> >
> > Q: Bigger model does not necessarily outperform a smaller one:
> >
> > A: Thanks for asking this fascinating question. While this statement may not be true in general nor can be proved theoretically (even in a non-private world), there is plenty of evidence that for NLP tasks that larger models tend to give better utility-vs-privacy tradeoffs [34, 74, X2].
> >
> >
> > Q: What is novel?
> >
> > A: Finally we address your question regarding novelty of the work. It is true that our work is empirical and does not have new mathematical results. Our goal was to bring to spotlight an important class of problems and algorithms related to model compression to the DP literature. Model compression is an extremely active area of research in non-private world ([43] for a survey), yet it has not received similar attention in the DP community. Case in point: there is not a single ICML, ICLR, NeurIPS paper on the topic. We believe that the model compression problem considered in our paper (where a single large model such as BERT is compressed into small BERT during fine-tuning stage) is new and different from the settings considered in other papers. Furthermore, our setting is more relevant to deployment of NLP models such as BERT, GPT2 etc. for common NLP tasks such as natural language understanding and next word prediction.
> >
> > From a technical standpoint, we believe that our paper shows some interesting results in DPIMP including its connections to the Lottery ticket hypothesis (we would love to hear your feedback on these sets of experiments since DPIMP provides better performance than DPKD). We believe that zero-shot initialization strategies for student models in DPKD is pretty surprising in its effectiveness in closing the gap. Given the importance of this problem, we think our work gives a substantial baseline for more work to follow.
> >
> > Finally, we believe that identifying a right and important problem, bringing the attention of the community towards solving it is in itself a worthy goal.
> >
> > We would very much appreciate the reviewer considering increasing their rating in case they find our responses compelling.
> >
> > X1: https://www.census.gov/programs-surveys/decennial-census/decade/2020/planning-management/process/disclosure-avoidance.html
> >
> > X2: Survey https://differentialprivacy.org/dp-fine-tuning/

---

> > > ### Comment · Reviewer_pc64 · 2022-08-04
> > > **Response to authors**
> > >
> > > I thank the authors for a detailed response. While I'm mostly satisfied with part 1 of their rebuttal, where they elaborated on the sections that were unclear to me, there are still some misconceptions left after the second part of the rebuttal. I additionally agree with most points addressed in the general response to authors. One thing that stands out to me still (some may claim that this is purely preferential however) is the argument on PATE being brought up because of the student-teacher terminology. I suggest if the authors strongly believe that their work should NOT be compared to PATE, then they should modify the terminology to alleviate any chance of confusion. Because evidently, as almost all reviewers picked up on it, so will the general audience.
> > >
> > > I just want to clarify that I am aware on the epsilon values used in other works, my question was specifically on the < 4.25 (as in where does the .25 come from).
> > >
> > > I fully agree that there are indeed certain interesting results presented in this paper and it is technically sound, however, the methodological contributions are still rather limited in my view. This being said, I believe that the main purpose of such works is to encourage the broader scientific community to participate in research not directly related to their own topic, which this paper tries to achieve. As a result, I am happy to marginally increase my score (from 4 to 5).

---

> > > > ### Author Response · Authors · 2022-08-04
> > > > **Response to Reviewer pc64**
> > > >
> > > > We thank the reviewer for their participation in the discussion. We appreciate your time, comments, and valuable suggestions.
> > > >
> > > > In hindsight, we agree with you that we should not have used the words teacher and student model but instead used Full-model and compressed model. This would make our paper more readable, and we intend to do that in the future versions. We would also include the formal problem statement. Thanks for your suggestions, and helping our work be more readable to the general audience.
> > > >
> > > > Finally, you write that there are still some misconceptions left after the second part of the rebuttal. We would be happy to clarify any questions you have.

---

### Official Review · Reviewer_BeSb · 2022-07-07

**Rating:** 6
**Confidence:** 5
**Soundness:** 3 good
**Presentation:** 3 good
**Contribution:** 3 good

**Summary:**

A commonly used technique in practical NLP systems is that of model distillation (or compression or sparsification). This technique converts a Large Language Model (LLM) into a more lightweight/small model so as to make the prediction faster and memory efficient.

On the other hand, Differentially Private (DP) training has recently started to become a standard notion of safety against various adversarial model attacks.

This work lies at the intersection of the above two concepts, in that it introduces methods to obtain compressed student models, from large teacher models, that have differential privacy guarantees. To that end, the paper investigates the following approaches:

1) Obtaining student models via a distillation loss function. Here DP is introduced possibly in the fine-tuning steps of teacher and students models.

2) Transforming teacher models into student models via Iterative Magnitude Pruning (IMP). Here DP is introduced in the fine-tuning steps.

**Questions:**

1) A main motivation for introducing DPKD is mentioned in lines 118-121. While that explanation with regards to the KD loss function makes intuitive sense, is it possible to formalize it properly? The text in those lines is used to justify the introduction of DPKD but it is quite hand-wavy and doesn't cite any previous work for support as well.

2) In Algorithm 1, line 3, what does it mean to initialize the student model with a privacy budget? How can just an initialization step have a privacy guarantee? This is also not clear from the experiments as there only the teacher and student model training steps involve DP.

3) What is the motivation for choosing privacy parameters, epsilon and delta, as shown in line 144? Calibration of these quantities is a problem in current literature as we don't know which values are good, so the current choice seems arbitrary.

4) As mentioned in the weaknesses, the DP guarantees for DPIMP methods are missing. Can we provide epsilon, delta values for the models resulting from those algorithms?

5) Related to the PATE works, there is also very recent work on introducing DP to LLMs at the prediction stage (see recent works such as https://arxiv.org/abs/2201.00971 and https://arxiv.org/abs/2205.13621). It will be helpful to add a few lines regarding these to improve the quality of presentation.

Minor typos:

1) Phrase "for initialization compressed models" in line 54.

2) "show" should be replaced with "shows" in line 55.

3) Phrase "opens a whole in direction" in line 295.

**Limitations:**

Yes.

**Strengths And Weaknesses:**

Strengths:

1) The setup and motivation is practically quite relevant. This is because both distillation and differential privacy are critical in large-scale NLP systems.

2) It studies two distinct compression methods: one based on the distillation loss function and the other based on weight pruning.

Weaknesses:

1) The DP guarantees for pruning based methods are entirely missing. For instance, the output of algorithm 2 is said to be a "private student model" but one doesn't know what the epsilon-privacy guarantee is. This makes the algorithm practically useless since the user doesn't have a calibration of privacy gains.

2) The experimental results are limited only to BERT and 1/2-BERT. It would be interesting to see if their results generalize to other architectures.

3) The technical contribution to introduce DP to model compression is a fairly straightforward application of existing tools and doesn't involve significant novelty.

---

> ### Author Response · Authors · 2022-08-02
> **Response to Reviewer BeSb**
>
> We thank you for a careful reading of the paper, positive feedback and questions. Below we address specific questions raised by you.
>
> Q: Motivation for introducing DPKD is mentioned in lines 118-121…
>
> A: Thanks for the question. Before we answer this question, let us recall the basics of analysis of DPSGD given in lines 87-91. In DPSGD, we add noise to clipped per-sample gradients and hence every iterate of DPSGD is private. In other words, after each iteration t of DPSGD, model weights W_t can be assumed to be public information. Now, in iteration t+1, gradients of samples are computed with respect to W_t, and hence privacy only depends on gradients belonging to the samples in a single batch. This is crucial for applying amplification by subsampling theorems in privacy analysis.
> Now consider a framework where the teacher models are trained using SGD on the dataset D and the student models are trained with DPSGD while minimizing the Equation 1 on dataset D. Such an algorithm does not output a differentially private compressed student model. This is due to the distillation loss term H(y_true, P_S) in Equation 1. Here, P_S is a function of the entire dataset as the teacher was not trained with DP. Therefore, gradients of samples are now functions of entire dataset D, which forbids us from applying subsampling theorems in privacy analysis. Our solution to circumvent this was to make P_S DP as well by training the teacher model with DP on dataset D.
>
> We hope this explanation helps. This is a subtle but important aspect of our DPKD algorithm. To the best of our knowledge, this is our contribution that does not appear in literature.
>
> Q: In Algorithm 1, line 3, what does it mean to initialize the student model with a privacy budget?
>
> A: Thanks for asking this clarifying question. You are right that in our experiments, the initialization of student model weights (in Section 3.4) does not incur any privacy cost, so \epsilon_2 = 0. Having said that, there could be student initialization strategies that are functions of the dataset D, in which case, we need to account for the privacy loss.  We will clarify this in the future versions of this paper.
>
> Q: Choosing privacy parameters, epsilon and delta, as shown in line 144?
>
> A: The privacy parameter \delta controls the failure to probability of not satisfying the DP guarantee. If N is the size of the private data set, it is good to have \delta << 1/N. The reason is simple: Consider a mechanism which randomly releases a sample from a dataset. Such an algorithm would satisfy DP with epsilon = 0 and \delta = 1/N. However, clearly this is not a good private algorithm. Text books recommend using \delta = 1/N^2, but all the recent works in deep learning with DP use \delta nearly 1/N. Compared to some recent works, our choice of delta = 1/10N is better.
> Our choice epsilon is again inspired by the recent works on deep learning with DP. Most of the papers in this literature use epsilon around 8. In this regard, we are using a smaller epsilon compared to the previous works, which means better privacy. Recently, we also did experiments with epsilon = 1, and obtained similar results. We included these results in Appendix B in the revised version.
>
> Q: Guarantees for Pruning are missing:
>
> A: We are sorry that you could not find the privacy guarantees of our DPIMP algorithm. We state them in Table 4 and 5, where we give the performance of compressed models produced by DPIMP. In fact, all the compressed models (using both DPIMP and DPKD) have a privacy value of epsilon < 4.25. If your question was how we allocate privacy budget across pruning iterations, then it is mentioned in the line 253-254. We allocate an equal privacy budget across all the iterations of DPIMP such that resulting epsilon < 4.25.

---

> > ### Author Response · Authors · 2022-08-02
> > **Response to Reviewer BeSb - part II**
> >
> > Q: Comment on recent work on DP at the prediction time.
> >
> > A: Thanks for the comment, and we added a few lines regarding these recent works of Majmudar et al. and Ginart et al. in the revised version. However, they study a different problem of protecting privacy during prediction time. While this is an interesting problem, our setting is quite different from this one. In our setting, we want to publish the student model to the public and want to ensure that no privacy violations occur. Observe that this is a stronger guarantee than DP at prediction time; a model satisfying our condition is automatically DP at prediction time but not vice versa. Consider a scenario where a company is trying to deploy a model trained on emails of several customers on the users’ phone. In this setting, it is important that weights of the network being installed on the users’ devices do not leak any privacy of the data on which it was trained. A model that is DP at the prediction time would not be able to protect against such attacks. We would like to note that most of the deep learning with DP literature considers the problem of releasing models privately.
> >
> > We would very much appreciate the reviewer considering increasing their rating in case they find our responses compelling.

---

> > > ### Comment · Reviewer_BeSb · 2022-08-06
> > > **Reviewer Response**
> > >
> > > I have two follow-up questions to the comments about privacy guarantees of DPIMP methods.
> > >
> > > 1) In the DPIMP methods, while the fine-tuning steps are performed with DP, there is are additional steps corresponding to pruning. If the pruning steps weren't present, typical composition calculations would certify the epsilon value of a resulting model obtained with DP-SGD. However, it is not clear, theoretically, how the epsilon value changes when the pruning steps are introduced.
> > >
> > > 2) In Algorithms 2 and 3, the list of inputs does not contain the epsilon parameter based on a desired privacy level. So then, experimentally how are tables 4 and 5 realized with epsilon values upper bounded by 4.25?
> > >
> > > Clarifications on the other points I raised are satisfactory.

---

> > > > ### Author Response · Authors · 2022-08-08
> > > > **Our answers.**
> > > >
> > > > Thanks for your time and involvement in the discussions.  We appreciate your questions. We are happy to answer your questions.
> > > >
> > > > 1. The pruning (steps 4,5,6 in Algorithm 2, steps 4 and 5 in Algorithm 3) do not incur any further privacy cost. To understand this better, let us recall the basics of analysis of DPSGD (which we outlined in the paper and also above in general reply to all reviewers).  In DPSGD analysis, it is crucial (to apply composition theorems) that all the model weights are released as public information after every iteration of the algorithm. In other words, after iteration t, the weights of model W_t are public information.  Further, we know that differential privacy satisfies post processing property. That is, if the output of an algorithm is DP, then any operation on the output without looking at the data will preserve DP with exactly same privacy cost.  Therefore, our proof that pruning steps (steps 4,5,6 in Algorithm 2,  steps 4 and 5 in Algorithm 3) do not incur any additional privacy loss simply follows from the post processing property of DP.   We will add a tiny theorem regarding this for clarity in the future versions.
> > > > On the other hand, if your question is on can we show any new privacy amplification theorems because we only retain a subset of weights is a very nice theoretical question. We leave it as an open question.
> > > >
> > > > 2. As we write in lines (253-254) and (276-277), we split the privacy budget equally among all the iterations of the DPSGD algorithm in Algorithms 2 and 3.
> > > >
> > > > 3. Finally, we did add a small discussion regarding variation in our epsilon values in the general rebuttal. We repeat it here for convenience. The reason epsilon values vary slightly across different GLUE tasks is due to the choice of \delta, which controls the failure probability of not satisfying DP guarantee. As we set \delta = 1/10N, our epsilon values also change as the privacy loss curve is a function of both epsilon, delta, and fixing one of them uniquely determines the other.

---

> > > > > ### Comment · Reviewer_BeSb · 2022-08-09
> > > > > **Reviewer Response**
> > > > >
> > > > > The authors have satisfactorily addressed all my comments, thanks for that. While I agree with their position on how this work brings light to the topic of model compression, after reading other reviews and comments, I also stand by my earlier comment of limited technical novelty. Therefore, I will maintain my score as it is.

---

### Official Review · Reviewer_yzKR · 2022-07-10

**Rating:** 5
**Confidence:** 4
**Soundness:** 2 fair
**Presentation:** 3 good
**Contribution:** 2 fair

**Summary:**

- The authors focused on compressing LLM in a differentially private way. Previous works mostly focused on differentially private fine-tuning but not the compression parts.
- The proposed method achieves very similar performance with 50% sparse model compression and guaranteeing differential privacy.
- The authors showed the limitation of straightforward ways of applying DPSGD to Knowledge Distillation and proposed the improved version of it to reduce the accuracy losses.

**Questions:**

1. Compression rate
- In the paper, the authors only show the experimental results with 50% sparsity.
- However, in many applications, we need much higher compression rates to increase the latency and reduce the costs.
- It would be great if the authors can provide more results with a diverse range of compression rates.

2. Epsilon value
- Based on the differential privacy definition (Section 2), epsilon determines the maximum differences between two probabilities with and without a single sample.
- If we select the epsilon value as 4 (which is < 4.25), the difference ratio is exp(4) = 55.
- In other words, the maximum probability differences would be significantly large.
- In this paper, most results are based on the epsilon value < 4.25. But I think this would be a too loose threshold.
- It would be good if the authors provide tighter privacy thresholds and the corresponding performances.

3. DistBERT
- Even though DistBERT is computationally expensive, we only need one time for training the DistBERT (because it is KD on public data).
- Then, we can use the DistBERT for various downstream tasks.
- In that case, I think the computational complexity of the DistBERT does not matter a lot.
- At the end, the objective is to achieve the differentially private compressed language models and the objectives seem the same between DistBERT and the proposed method.
- In that point of view, the advantages of the proposed method in comparison to DistBERT are limited.

4. Structured DPIMP vs Unstructured DPIMP
- It would be good to show the quantitative inference latency comparison between these two methods.
- Also, directly comparing between unstructured DPIMP and DistBERT is not the fair way.
- Actually, based on Table 4, DistBERT seems consistently better than Structured DPIMP.

**Limitations:**

- As the authors said in the introduction, the proposed method is not limited to the NLP.
- In that case, it would be good to show how well those compression methods (with DP) can be applicable to other domains in the appendix.
- Also, the authors do not consider the PATE framework for model compression. However, in some applications, PATE works better than DP-SGD. To show the generalization of the proposed method, it would be better to discuss and consider PATE as well.

**Strengths And Weaknesses:**

Strength:
- The authors tackled an important but under-explored problem: DP for model compression in LLM.
- The paper is well-written and easy to follow.

Weakness:
- The experiments are limited. It would be good if the authors provide more diverse experiments with varying compression rates.
- Epsilon value is set to be too high. With epsilon < 4.25, I am not sure whether the achieved model is private enough.
- The advantages from DistBERT is not convincing. Comparing with unstructured DPIMP is not fair.

---

> ### Author Response · Authors · 2022-08-02
> **Response to Reviewer yzKR**
>
> We thank you for a detailed review of our paper, giving us positive feedback, and fair comments. Below we will address specific questions you asked.
>
> Epsilon Value. We agree that epsilon around 4 used in our work may seem to offer low privacy protection at the first glance. However, it is important to keep in mind that this is a worst-case guarantee, where it is assumed that the adversary knows all data except the one we are protecting the privacy of. So actual protection on real datasets can be significantly higher. Indeed, many works have shown that even with epsilon = 8, one enjoys significant protection against membership inference attacks; for example [76] show that membership inference attacks on BERT models trained with epsilon = 8 is no better than random guessing (50% success rate). Moreover, almost all previous works in deep learning space have used epsilon values in this range; see the SOTA papers in NLP and images recognition tasks published last year [34, 74, 12, 42]; most of those papers use epsilon in the range of [4-8]. Moreover, the US census bureau, which is the largest deployment of DP used epsilon > 18 [X1]. Considering everything, we believe that our choice of epsilon offers more privacy protection.
>
> Experiments for epsilon value: Based on the reviewer’s suggestion, we repeated all of our experiments with a strict privacy budget epsilon = 1 and presented the results in Appendix B in the revised version. The observations from the original experiments continue to hold in this regime, and in particular relative drop in accuracy between large models and compressed models remains roughly the same.
>
> DistillBERT: Thanks for drawing attention to the performance of DistillBERT. The advantages/disadvantages of our proposed methods over distillBERT is a worthy point to discuss more. As you can see from our experiments, only unstructured IMP beats the performance of distillBERT, whereas structured IMP and zero-shot initialization strategies come close. Understanding this gap was precisely our goal: quantify how much performance one loses if we do not have access to pre-trained small student models such as distillBERT and how we can mitigate it. But as you also alluded, access to pretrained student models is not always an option. For example, suppose an application wants 1/15th of BERT; where would we find such a pretrained model? One option is to repeat the pretraining algorithm of distillBERT to produce 1/15-BERT. However, there are plenty of scenarios where this is not an option either due to infrastructure issues or due to cost or both. Our goal was to provide algorithms for model compression in such scenarios for fine-tuning stages that can match the performance of distilBERT. One way to interpret our results is to say that, if one can pretrain student models, then it could lead to better performance; if not, one could use the strategies described in our work and get an estimate of how much performance is left on the table. We believe this is a valuable information as pretraining costs for LLMs (even 1/10 of BERT has millions of parameters) is prohibitively large.
>
> Compression Factor: The above discussion also reveals why we chose the compression ratio of 50% for our experiments. We wanted to compare against distillBERT, which is a widely available public pretrained compressed model. However, given a chance, we would conduct experiments to compare against pretrained models with lower compression ratio.
>
> Structured-vs-Unstructured Compression: This is a great point, and we have acknowledged already that sparsity alone is not an accurate measure of inference latency. However, sparsity is one of the widely accepted measures of model compression starting from the pioneering works Lecun et al of [32] to the recent award winning work on Lottery Ticket Hypothesis [20, 11].
>
> Experiments on Image classification. While our algorithms are not specific to a particular domain, our focus was on NLP applications. We hope that you agree that this is an important domain and worthy of full attention. We expect that our work will trigger more research in other domains including image classification, multimodal models, etc.
>
> PATE framework for model compression. Please see our general comment for all reviewers for a detailed discussion on this point.
>
> We would very much appreciate the reviewer considering increasing their rating in case they find our responses compelling.
>
> X1: https://www.census.gov/programs-surveys/decennial-census/decade/2020/planning-management/process/disclosure-avoidance.html

---

> > ### Comment · Reviewer_yzKR · 2022-08-08
> > **Thank you for your rebuttals**
> >
> > Thank you for the rebuttals. I carefully read other reviews and the general responses from the authors as well.
> >
> > First, thank you for providing the experimental results with epsilon = 1.0. As I expected, the performance drops from epsilon = 4.25 to epsilon = 1.0 is significant.
> > Also, I think the justification in comparison to DistillBERT is still not valid. With enough computational costs (one time only), I am not sure why the proposed method is superior to DIstillBERT. In that point of view, I am standing to my original score (4). But I will not be upset if the AC recommends acceptance for this paper.
> >
> > 1. Epsilon
> > - I understand that other works also use the large epsilon values.
> > - However, it cannot be a backup that this work can also use large epsilon values.
> > - The main advantage of DP is that DP can guarantee something theoretically. However, if that guarantee is somewhat useless (like epsilon = exp(4.25)), there is no point to using DP at all.
> > - As you said, it would be better to use membership inference attack as the privacy metrics instead of epsilon in DP.
> > - But thank you for providing the results with epsilon = exp(1). And the performance drop from epsilon = exp(4.25) seems significant.
> >
> > 2. DistillBERT
> > - I understand your points on DistillBERT for general compression ratios.
> > - But I think my point is still the same that we only need "one-time" training for any size of DistillBERT.
> > - In that case, do the computational costs matter a lot?
> >
> > 3. Compression rates
> > - The main reason that I want to see the results with various compression rates is to check whether this method can be generalized to any compression sizes.
> > - If this can be shown, the authors can claim the advantages from DistillBERT better.- Unfortunately, this paper does not have this result.

---

> > > ### Author Response · Authors · 2022-08-08
> > > **Thanks for your questions.**
> > >
> > > Thanks for your time and involvement in the discussions.  We appreciate your questions.
> > >
> > > 1. Epsilon. We agree with you that smaller values of epsilon are better, and as a community we are striving hard to train state of the art deep learning models with epsilon << 1. We are not there yet, and we will continue to push the frontiers.  Moreover, what is a good value of epsilon is perennial topic of discussion ever since DP models started to be deployed, so, we will focus on something more specific here.
> > >
> > > We believe that some additional comments regarding worst case analysis and membership inference attacks may help. We want to reword our responses in the rebuttal concerning this comment.
> > >
> > > The current theoretical analysis shows that worst case privacy loss is around 4. This does not mean that even theoretically privacy loss is 4; it is an upper bound. It could be far lower than that but we are unable to prove it due to various mathematical reasons. On the other hand, some of works on membership inference attacks show that models trained with epsilon = 8 offer significant privacy protection, and adversary's chances of success  is as low as random guessing, which translates to very small values of epsilon.
> > >
> > > Thus, our reported epsilon = 4 is only an upper bound. In our experience having an upper bound  which has a mathematical proof  is crucial in many deployment scenarios as empirical tools (such as membership inference attacks) are specific to datasets.  Thus, to us, getting rid of reporting (large) epsilon values which come with mathematical proofs and completely replacing them with empirical hypothesis testing style experiments can lead to privacy solutions that are not well grounded and come with no concrete guarantees.
> > >
> > > 2. DistillBERT.  We thank you for your comments. The cost of pre-training  can be prohibitively large even if it is only one time. Moreover, it is not merely about cost of pre-training, but also also about infrastructure, pretraining datasets, etc. Our algorithms cater to scenarios where pre-training compressed models is hard. We believe that this is an important scenario to study.
> > >
> > > 3. Compression Rates. We will try to do these experiments for future revisions of the paper.

---

> > > > ### Comment · Reviewer_yzKR · 2022-08-09
> > > > **Thank you for the reply**
> > > >
> > > > 1. Epsilon
> > > > - I agreed that this epsilon problem is not only the problem of this paper. It is the problem of DP itself.
> > > > - However, what the authors claimed in this rebuttal does not make sense.
> > > > - As the authors said, theoretical guarantee is important. However, what I want to say is that whether the theoretical guarantee with epsilon = 4.25 is meaningful? With epsilon = 4.25, what we can theoretically guarantee is that the upper bound of the probability difference ratio with and without one sample is exp(4.25) = 70. I do not think this is a useful guarantee at all.
> > > > - In addition, as you said, the membership inference attack metrics are specific to datasets. In that case, with epsilon = 8, the claim that membership inference attack metrics are very small is also data specific claim.
> > > > - The logic that the authors provide in this rebuttal has many counter-examples; thus, those responses do not make me convince.
> > > >
> > > > 2. DistillBERT
> > > > - I understand that there are some advantages of the proposed method in comparison to DistillBERT.
> > > > - However, those are not well explained in both rebuttals and current manuscript. Please revise the manuscript that can clearly compare the advantages and disadvantages of the proposed method in comparison to the DistillBERT.
> > > >
> > > > I am going to increase my score to 5 but it is definitely the maximum score that I can provide.
> > > > If this paper is accepted, it would be great if the authors can address all the comments about the epsilon, DistillBERT and various compression ratios.
> > > >
> > > > Thank you for the hard working on this rebuttal.

---

> > > > > ### Author Response · Authors · 2022-08-09
> > > > > **Thank you for discussions**
> > > > >
> > > > > We thank you for participating in the discussions.  We will include experiments with epsilon = 1 in the future revisions of the paper.  We will add more discussions on pros/cons of our approach DistillBERT (or any pre-trained compressed models) and elaborate on where our work is applicable. We appreciate your time and feedback on our paper.

---

### Official Review · Reviewer_D2Eo · 2022-07-11

**Rating:** 5
**Confidence:** 4
**Soundness:** 3 good
**Presentation:** 3 good
**Contribution:** 2 fair

**Summary:**

This paper focus on a important scenario to consider differential privacy and model compression together. It considers KD and (structured and unstructed) pruning. The paper is generally well-written.

**Questions:**

In the abstract, the authors state "we initiate the study of differentially private model compression and propose frameworks for achieving 50% sparsity levels while **maintaining nearly full performance**". From Tab. 2/3/4/5, it seems it does not achieve as well as the authors claim.  A relatively better result is achieved by unstructured DPIMP, which has 83.5 vs. 85.2 (raw BERT) in AVG (there is still a tiny gap); however, unstructured pruning with 50% sparsity usually has a limited speed-up during inferencing. The above statement seems slightly overclaimed.

The paper seems to report relatively weak results in GLUE benchmark, not only for the proposed method but also for existing work. By checking other papers (e.g. DistilBERT https://arxiv.org/pdf/1910.01108.pdf or even a stronger one  TinyBERT  https://arxiv.org/abs/1909.10351), we could observe that the reported results for BERT and DistilBERT are lower than original papers. It is unclear whether the results are evaluated in DEV or TEST set. Could you please provide more details? Frankly speaking, with half the size of BERT-base parameters, such achieved performance might not be competitive with some models without pretraining (which might be much faster).

**Strengths And Weaknesses:**

Strengths:
- it is interesting to consider DP and compression together.
- consider both KD and (structured and unstructed)  pruning.
- the paper is well-written

Weaknesses:
- some statements are slightly over-claimed
- the methods seems not that novel. For example, DPKD is like a pipline to bring many existed methods together.
- Sec. 3.4 and 3.5 do not introduce anything beneficial, but something that has nearly idential results -- strategies to make better students/teachers.

---

> ### Author Response · Authors · 2022-08-02
> **Response to Reviewer D2Eo**
>
> We thank the reviewer for positive feedback, comments, and questions. We address the specific questions asked by you.
>
> Q: some statements are slightly over-claimed.
>
> A:  We are sorry that you think we overclaimed our results. We respect your opinion and would be happy to reword our sentences according to your suggestions. For our future reference, we would like to understand which part of the sentence you felt is overclaimed. First note that we use the word sparsity instead of inference latency in abstract and our contributions. We did not claim that our models have 50% shorter inference time nowhere in our submission. Moreover, in lines 228-231, we discuss sparsity and inference latency, and we explicitly point out that sparisity does not correlate with inference latency. However, many works in model compression literature, including the recent award winning works of [20, 11] and seminal work of [32], use sparsity as the measure of model compression. Further, their motivation to study sparsity is also to minimize inference latency and energy costs. To our understanding, a part of the reason to use sparsity as a measure of model compression is that measuring inference latency of models depends on the hardware configurations and it is not easy. Sparsity gives a more clear and abstract measure of model complexity, besides being interesting on its own as a tool for measuring function representations. Second, your comment seems to imply that our compressed models performance is significantly below the performance of  large language models, and hence we should not use the word “nearly”. We are happy to replace nearly with 1.7%, as our DPIMP obtains sparsity of 50% and is only 1.7% below the performance of full BERT (Table 5).
>
> Would you agree if we replace our original sentence with the following: we initiate the study of differentially private model compression and propose frameworks for achieving 50% sparsity levels while guaranteeing that performance drop compared to full model is small; for some of our algorithms average performance drop on GLUE benchmark is 1.7%.
>
> Q: The paper seems to report relatively weak results in GLUE benchmark.
>
> A:  Thanks for your question. We think there is a small misunderstanding in reading our tables. We are comparing performance of full BERT models trained with exactly the same privacy parameters to that of compressed models with 50% sparsity. We are not comparing performance of our compressed models to non-private BERT models, which is what you are referring to in those citations. DP guarantee already comes with some performance drop, which is well documented in prior works. In this work, we are studying the relative performance drop of private models when we impose model sparsity constraints. We hope this answers your question.
>
> Q. Section 3.4 and 3.5 do not introduce anything beneficial.
>
> A: Thanks for the question. In Section 3.4 we propose zero-shot initialization strategies, and Table 2 shows that this gives nearly 18% boost to average performance compared to random initialization (Table 1). In section 3.5 we address the question, can larger models, which are known to achieve better privacy-vs-utility tradeoffs [34, 74, 12, 42], be better teachers. Unfortunately, the answer turns out to be no, but we found this counter intuitive. While it is fair to say that Section 3.5 does not improve the results over section 3.4, it brings to light an interesting phenomenon that larger models, which achieve better utility, are not necessarily better teachers for model compression in the DP world.
>
> We would very much appreciate the reviewer considering increasing their rating in case they find our responses compelling.

---

> > ### Comment · Reviewer_D2Eo · 2022-08-09
> > **Thanks for your replies**
> >
> > Q1: It is quite subjective to say if a performance drop of1.7% is small or negligible. But, anyway, the paper is not on the focus of performance. It would be nice to state something in a objective manner.
> >
> > Q2: Thanks for your explaination. To better contexualize your result, it would be nice if you could also show the raw BERT/DistilBERT result.
> > How about the comment "it is unclear whether the results in GLUE are evaluated in DEV or TEST set"
> >
> > Q3: Thanks, the counter intuitive result is indeed interesting.

---

> > > ### Author Response · Authors · 2022-08-09
> > > **Response to Reviewer D2Eo**
> > >
> > > Thanks for your time and involvement in the discussions. Regarding Q2, we follow the methodology in the literature and evaluate the results in GLUE using the DEV set.

---

### Author Response · Authors · 2022-08-02
**General comment for all reviewers - part II continued**

Now we are ready to state why we did not consider the PATE framework to solve our problem. We give 3 main reasons along 3 different axes.

1) Technical:  A natural way of using the PATE framework in our setting would be to partition the sensitive dataset D into say 10 teacher models M_A1, M_A2… M_A10, where the architecture of each teacher model is M_A. We fine-tune each teacher model M_Ai on the portion of the sensitive data set D. The student model M_B would aggregate the predictions of the teacher models to learn. However, to do private aggregation of the teacher ensemble (hence the name PATE), the student model would require access to a non-private dataset D’ which has the same distribution as D [47, 48]. In other words, if D is a MNLI dataset, we would require additional data which has a similar distribution to MNLI dataset. We do not have access to such a non-private dataset D’ in our problem statement.
Alternatively, we can use the test set of D (i.e, test set of MNLI) as the public data set for student training. However, this would lead to unfair comparisons with previous approaches and comparing our results to the literature. Doing so also has the following drawback noted in the original paper [47]: “Note that this may increase the variance of our test set accuracy measurements, when compared to those computed over the entire test data.”

2) No Published Work on Solving GLUE via PATE: Ensembling learning is a method for training multiple ML models (often using different algorithms) to obtain better predictive performance, and PATE provides a framework for DP aggregating these predictions to train a new model. Thus, PATE is a framework for training any ML model, and should not be confused as a model compression technique akin to pruning. It is true that ensemble learning framework can be applied for model compression but that is not the focus of this paper; see next point. The performance of PATE for training large deep learning models lags behind the training via DPSGD, and is well known in the community. To provide evidence, SOTA results in NLP and image classification from the past year all use DPSGD [34, 74, 76, 12, 42].
We do not know of any paper that studies PATE framework for solving GLUE benchmarks using Large Language Models (LLMs). Our DPIMP framework achieves average accuracy of ​​83.5% for GLUE benchmark using half as many parameters as BERT. We do not know of any published work that achieves an accuracy of greater than 80% on GLUE benchmark using PATE framework using any BERT model.

3) Model Compression in Private vs Non-private World: The focus of this work was to study the most widely used compression algorithms in the non-private world based on our experience and understanding, and make them differentially private and evaluate their performance. Thus, we chose Iterative Magnitude Pruning and SGD with KD objective function. Based on our experience and understanding these two most widely used algorithms for model compression; see also this survey [43]. Also observe that ensemble learning is not a dominant method of model compression in the non-private world, as evident from these highly cited papers [51, 31].

Having said that, we are not making any claims that our work is exhaustive. It is true that PATE can be applied for model compression, but there are many other techniques such as quantization, data augmentation, etc., that can also be applied; see for example this excellent survey on the topic [43]. However, doing such an exhaustive comparison in a single paper is beyond scope. We believe that part of the reason why some of the DP papers think PATE as a natural framework for model compression is partly due to the nomenclature; both compression literature and PATE use the words teacher and student models. But it is important to note that they do not have the same semantic meaning. Moreover, the frameworks for compression algorithms studied in this paper themselves are important enough that they deserve full attention, and we wanted to be as comprehensive as we can in the choices we have made, including as the reviewer pc64 said pointing out techniques that do not work.

We hope that our paper paves way for more research in DP model compression, exploring more algorithms and new theoretical analysis, perhaps even improving our results. Given the significance of this problem both in theory and practice, it would benefit the entire community.

---

### Author Response · Authors · 2022-08-02
**General comment for all reviewers - part I**

In this comment, we want to give updates about new experiments, add some more details to our problem description and our frameworks.

New experiments

We repeated all of our experiments with a smaller privacy budget of epsilon = 1 and presented the results in Appendix B in the revised version. The observations from the original experiments continue to hold in this regime. In particular, the relative drop in accuracy between large models and compressed models using our frameworks remains roughly the same.

Model Compression, Ensemble Learning, PATE, and overloading of words student and teacher models.

There seems to be some misunderstanding regarding the model compression problem considered in this work, ensemble learning, and PATE framework. This confusion has resulted in drawing some unfair comparisons between our results and the PATE framework; to us, it appears, this confusion is primarily due to usage of terms “teacher” and “student” models in both compression literature and PATE framework. In a nutshell, we are not considering the ensemble learning framework, and hence there is no necessity in our frameworks to do private aggregation (PATE). Our focus was to study the most dominant model compression algorithms used in practice (in the non-private setting) and how to incorporate DP constraints. Although we addressed this in lines 68-80, here we would like to further elaborate on this. To make everything precise, let us formally state our problem, which we illustrated in Figure 1. We will include this formal problem statement in the full version for more clarity.

Problem Statement: Input to our problem is privacy parameters (\epsilon, \delta), a large model M_A with initial model parameters \theta_A(0), a private sensitive dataset D from a downstream task which we want to solve, and a compression factor \gamma.  Let |M_A| denote the parameter count of M_A. Our goal is to produce a compressed model M_B satisfying two constraints:
1) |M_B| \leq \gamma \cdot  |M_A|.
2) The final weights of model M_B (denoted by \theta_B(t)) should be (\epsilon, \delta)-differentially private with respect to dataset D. A compression algorithm can make use of M_A in an arbitrary way as long the final weights of model M_B (\theta_B(t)) are differentially private with respect to dataset D.

Accuracy Comparisons: We measure the quality of compression algorithms by comparing the accuracy obtained by M_B satisfying (\epsilon, \delta)-DP on downstream task D to the accuracy obtained by M_A satisfying (\epsilon, \delta)-DP on downstream task D. This allows us to quantify how much performance one loses in private training due to model compression. Note that we are not comparing against the performance of non-private models.  We would like to find compression algorithms where differentially private M_B has the nearly the same performance of differentially private M_A.

No New Assumptions: In this work, we assume that M_A is BERT with initial parameters \theta_A(0) obtained by pretraining. We do not have access to pretraining dataset and we do not enforce DP on the pretraining step. The only dataset available for compression algorithms are sensitive datasets D and nothing else. This is exactly the model assumed in almost all DP-NLP papers such as [34, 74, 76], and we are not making any new assumptions compared to the previous works. Our choice of datasets from GLUE benchmarks are also inspired by the recent works in DP-NLP [34, 74, 76]. Given this problem statement, our results show performance of two compression algorithms DPKD and DPIMP while satisfying (\epsilon, \delta)-DP for \epsilon in the range of [4, 4.25], and \delta = 1/10N, where N is the size of the dataset, when compression factor \gamma = 1/2. All our results satisfy DP guarantees.

Slight Variation in Epsilon Values:  The reason epsilon values vary slightly across different GLUE tasks is due to choice of \delta, which controls the failure probability of not satisfying DP guarantee. As we set \delta = 1/10N, our epsilon values also change as the privacy loss curve is a function of both epsilon, delta, and fixing one of them uniquely determines the other; See [23] more details.

---

### Meta-Review · Area_Chair_EGWo · 2022-08-27

**Recommendation:** Accept
**Confidence:** Certain

**Metareview:**

This work proposes and empirically evaluates algorithms for compressing and fine-tuning a large model for a downstream task, while satisfying DP for the downstream task training data. The set up is the following: we have a large pre-trained language model such as BERT. We would like fine-tune it for a task using a dataset D, as well as compress it to a smaller model. The paper studies algorithms that are DP with respect to D and do fine-tuning+compression. The authors propose and evaluate different strategies for this problem and compare the privacy-utility tradeoffs.
The reviewers found the empirical evaluation to be thorough. Some of the other concerns raised by the reviewers have been addressed to my (and in most cases, their) satisfaction.
I think the problem studied by the paper is timely and important. I view the paper largely as a solid empirical study of natural algorithms for this problem. While the paper can be improved as discussed in the reviews and rebuttal, I believe it brings attention to an important problem and makes solid progress on it. I would therefore recommend acceptance.

**Award:**

No

---

### Decision · Program_Chairs · 2022-09-14

Accept